# SPECTRAL CONTRASTIVE REGRESSION

## ABSTRACT

While several techniques have been proposed to enhance the generalization of deep learning models for classification problems, limited research has been conducted on improving generalization for regression tasks. This is primarily due to the continuous nature of regression labels, which makes it challenging to directly apply classification-based techniques to regression tasks. Conversely, existing regression methods overlook feature-level generalization and primarily focus on data augmentation using linear interpolation, which may not be an effective approach for synthesizing data for regression. In this paper, we introduce a novel generalization method for regression tasks based on the metric learning assumption that the distance between features and labels should be proportional. Unlike previous approaches that solely consider the scale prediction of this proportion and disregard its variation among samples, we argue that this proportion is not constant and can be defined as a mapping function. Additionally, we propose minimizing the error of this function and stabilizing its fluctuating behavior by smoothing out its variations. The t-SNE visualization of the embedding space demonstrates that our proposed loss function generates a more discriminative pattern with reduced variance. To enhance Out-of-Distribution (OOD) generalization, we leverage the characteristics of the spectral norm (*i.e.,* the sub-multiplicativity of the spectral norm of the feature matrix can be expressed as Frobenius norm of the output), and align the maximum singular value of the feature matrices across different domains. Experimental results on the MPI3D benchmark dataset reveal that aligning the spectral norms significantly improves the unstable performance on OOD data. We conduct experiments on eight benchmark datasets for domain generalization in regression, and our method consistently outperforms state-of-the-art approaches in the majority of cases. Our code is available in an anonymous repository, and it will be made publicly available upon acceptance of the paper: https://github.com/workerasd/SCR.

## 1 INTRODUCTION

Continuous label prediction, known as regression, is widely utilized across various domains, including computer vision (Zhang *et al.*, 2015; Chen *et al.*, 2016), medical testing (Gilsanz & Ratib, 2011; Agatston *et al.*, 1990), and financial analysis (Happersberger, 2021). Unlike classification, which seeks to determine optimal decision boundaries, regression involves fitting outputs to a continuous function (Lee & Landgrebe, 1993). Therefore, when addressing challenges such as uncertainty estimation (Hüllermeier & Waegeman, 2021) and generalization (Yao *et al.*, 2022) in regression, it is crucial to consider the relationships between the labels.

While out-of-distribution generalization has received significant attention for classification (Wang *et al.*, 2022), regression generalization remains relatively underexplored. Particularly, the existing representation learning based methods like IRM (Arjovsky *et al.*, 2019) are primarily designed for classification tasks. The augmentation-based approach of C-Mixup (Yao *et al.*, 2022) has recently been proposed to enhance generalization by mixing training pairs based on the probability associated with label distances. While the aforementioned approaches are applied or can be adapted for regression generalization, their performance is limited because they do not account for the contrastive interdependence between features and labels.

To tackle the aforementioned problem and with the aim of learning a generalizable representation from the source domains, we introduce a contrastive learning loss specifically designed for regres-

sion. This loss brings features with smaller label distances closer together in the learned representation, while simultaneously pushing features with larger label distances farther apart, ultimately helping to separate representations learned from different domains and enhancing the generalization performance in the target domain. Contrary to the assumption in Regression Metric Loss (RML) (Chao *et al.*, 2022) that the ratio between feature distance and label distance is constant, we propose that this ratio varies and only equals a constant under certain ideal conditions. We argue that RML, by overlooking the variability in this ratio, may obscure the pattern of feature distributions in certain cases, as demonstrated in our experiments.

Specifically, motivated by augmentation-based techniques (Xu *et al.*, 2021; Sicilia *et al.*, 2023; Yao *et al.*, 2022) for domain generalization in regression and classification, we propose to generate new distributions by mixing pairs of training data. For each distribution, we create a metric penalty to identify discriminative patterns within the feature distribution. We align the real and synthesized distributions by minimizing the difference between the spectral norms of their feature representations. With the property of spectral norm, the minimization keeps the output scale from standing out, while lowering the upper bound of distribution discrepancy in regression.

The main contributions of this paper are three-folded:

1. Unlike prior methods that treat the feature-label distance proportion as fixed, we propose to model this as a variable mapping function and address the instability arising from fluctuations in this mapping.

2. To improve the OOD generalization, we expand the training distribution by generating new samples Yao *et al.* (2022), and then align the real and synthesized distributions by minimizing the difference between the spectral norm of their feature representations.

3. We conduct experiments on eight benchmark regression datasets and show that our method outperforms the state-of-the-art in most cases. The t-SNE visualization of the feature embedding illustrates the effectiveness and stability of our proposed metric loss.

## 2 RELATED WORK

### 2.1 METRIC LEARNING

Metric learning has been shown to be effective when related to methods that rely on distances and similarities (Kulis *et al.*, 2013). Traditionally, methods like PCA (Pearson, 1901) and KNN are widely used in the area of machine learning. With the development of deep learning, networks (Schroff *et al.*, 2015; Bromley *et al.*, 1993) related to pair distances are designed to correlate among samples while using shared weights in deep learning (Kaya & Bilge, 2019). Then, prototype-based metric losses (Wen *et al.*, 2016; Deng *et al.*, 2019) were proposed based on contrastive motivation. In regression tasks, the metric learning loss has not been well-defined because it is hard to build the connection between the metric distance and continuous labels. Recently, Chao *et al.* (2022) proposed an assumption that there is a constant proportion between the feature distance and the label distance. However, the method based on this assumption only considers the scale of the feature matrix, ignoring fluctuations in the proportion map. To solve this issue, this paper assumes that the proportion is a mapping function in the training process and proposes a metric loss to smooth fluctuations.

### 2.2 OUT-OF-DISTRIBUTION GENERALIZATION

Out-of-distribution (OOD) generalization aims at generalizing the model from the training distribution to an unseen distribution. Mostly, the methods can be divided into 3 parts (Wang *et al.*, 2022): data augmentation, representation learning, and training strategy. Data augmentation methods (Zhang *et al.*, 2018; Zhou *et al.*, 2021) utilize linear interpolation to fill the distribution gap, and some methods (Xu *et al.*, 2021; Sicilia *et al.*, 2023) also generate a new distribution to enrich the convex hull supported by the source distributions. Representation learning (Arjovsky *et al.*, 2019; Albuquerque *et al.*, 2019) aims at generating distribution-invariant feature representations from source distributions. Recently, methods like SWAD (Cha *et al.*, 2021) proposed some novel training and model selection strategies, significantly improving performance in OOD generalization.

## 2.3 GENERALIZATION IN REGRESSION

Recent research targeting generalization in regression tasks is based on data augmentation in which mixup pairs are selected based on the probability related to label distances (Yao *et al.*, 2022; Yang *et al.*, 2021). Even though limited research has been proposed on this topic, some methods designed for regression tasks can be transferred to generalization purposes. For instance, due to the function of metric learning, the metric loss in regression (Chao *et al.*, 2022; Gong *et al.*, 2022) can be regarded as an in-distribution generalization method. Also, distribution alignment methods in regression (Nejjar *et al.*, 2023; Chen *et al.*, 2021) can be updated as OOD generalization methods. However, these distribution alignment methods are not related to the label functions, which are supposed to be very important in regression tasks.

## 3 METHODOLOGY

### 3.1 PROBLEM DEFINITION

**Regression in deep learning.** Let $\{(x_i, y_i)\}_{i=1}^N$ be the dataset with $N$ samples, with $x_i \in \mathcal{X}$ being the input sample $i \in \mathbb{R}^+$ and $y_i \in \mathcal{Y}$ its corresponding label, and $\mathcal{X}$ and $\mathcal{Y}$ denoting the input space and the continuous label space, respectively. In the training phase, the network learns a projection function $g : \mathcal{X} \rightarrow \mathcal{F}$ and a regression function $p : \mathcal{F} \rightarrow \mathcal{Y}$. The projection function $g$ transforms the input data into the feature space, and the regression function $p$ maps the compact feature representation to the label space. The objective of the regressor is to bring the output prediction $\hat{y}_i$ close to the ground truth label $y_i$. Ideally, the optimal predictor $p$ is a fully connected layer that satisfies $y_i = \hat{y}_i = W_p^* f_i + b_p^*$, where $f_i = g(x_i)$ is the extracted feature, $W_p^*$ is the optimal weight, and $b_p^*$ is the optimal bias.

**Distribution discrepancy in regression.** Cortes & Mohri (2011) defines a theory of learning from different distributions in regression. Given the hypothesis $h$ being a map from input space $\mathcal{X}$ to the label space $\mathcal{Y}$, the discrepancy distance $disc$ between two distributions $P$ and $Q$ is defined as:

$$disc(P, Q) = \max_{h, h' \in H} |\mathcal{L}_P(h', h) - \mathcal{L}_Q(h', h)|$$

Here, the hypothesis $H$ is a subspace of the reproducing kernel Hilbert space (RKHS) $\mathbb{H}$ and $\mathcal{L}_D(h', h) = E_{x \sim D}[L(h(x), h'(x))]$, with $L$ being a MSE loss.

### 3.2 RELATIONAL CONTRASTIVE LEARNING

Prior works show that by leveraging the discrete labels to define positive and negative pairs in classification models, contrastive learning aims to learn feature representations with low intra-class variance and high inter-class separation, which can improve the generalization ability of the learned model. However, this motivation is based on the fact that the labels are discrete. In regression tasks, given an input-label pair of $(x_i, y_i)$, $\forall \epsilon > 0$, with input $x_{i+\epsilon}$ and its continuous label $y_{i+\epsilon}$, it's proven that $p$ should be a continuous bijection (Chao *et al.*, 2022), with homeomorphic label and feature distributions. Intuitively, there is a positive relationship between the distances of labels and distances of features - as the distance between two labels increases, the distance between their corresponding features should also increase, meaning that when two examples have labels that are farther apart, their representations in feature space should also be farther apart, and vice versa for labels that are closer together.

**Remark 1.** $d(y_i, y_j) < d(y_i, y_k) \iff d(f_i, f_j) < d(f_i, f_k), \forall i, j, k \in \mathbb{R}^+$

Note that, for any bounded open subset in $\mathcal{F}$, $p$ should be convergent and bounded, which means $p$ should be uniformly continuous on any bounded open subset (Rudin, 1976). Then, Remark 1 should be updated.

**Remark 2.** $d(y_i, y_j) < d(y_t, y_k) \iff d(f_i, f_j) < d(f_t, f_k), \forall i, j, k, t \in \mathbb{R}^+$

Remark 2 is not trivial. Since $\mathcal{F}$ is a compact space and label $\mathcal{Y}$ is continuous, then for $\forall \epsilon > 0$, we can find labels $y', y''$ with $d(y', y'') = \epsilon$. Then, $\exists \delta = d(f', f'') > 0$, such that $\forall d(f_a, f_b) < \delta$, we have $d(y_a, y_b) < \epsilon$. So, Remark 2 keeps $p$ uniformly continuous.

In light of the discussion above, we argue that the distance between labels can not be ignored in the regression tasks. In particular, we propose learning a feature-label proportional distance instead of the traditional distance, *e.g.* Euclidean distance between features:

$$d_r(f_i, f_j) = \frac{d(f_i, f_j)}{d(y_i, y_j)}, \tag{1}$$

Here, $d(\cdot, \cdot)$ represents Euclidean distance and $d_r(\cdot, \cdot)$ denotes the proportional distance induced from $d(\cdot, \cdot)$. In addition, $d_r(\cdot, \cdot)$ should be a bounded distance, which can be illustrated by the following theorem.

**Theorem 1.** *Given any two data points $(x_i, y_i)$ and $(x_j, y_j)$, we have $\|f_i - f_j\|_p \leq \|W_p^{*-1}\|_p \|y_i - y_j\|_p$. Here, $W_p^*$ is the optimal weight of the fully connected layer. $f_i, f_j$ are the features extracted from $x_i, x_j$ through model $g$, and $\|\cdot\|_p$ is the norm under $L_p$ space.*

**Proof 1.** *Given the optimal weight $W_p^*$, bias $b_p^*$ and data $(x_i, y_i)$, $(x_j, y_j)$, we have*

$$y_i = W_p^* f_i + b_p^*, y_j = W_p^* f_j + b_p^*$$

*where $f_i, f_j$ are extracted features from $x_i, x_j$, respectively. Then,*

$$\|f_i - f_j\|_p = \|W_p^{*-1}(y_i - y_j)\|_p \leq \|W_p^{*-1}\|_p \|y_i - y_j\|_p$$

Theorem 1 gives the upper bound of $d_r(\cdot, \cdot)$ which is $\|W_p^{*-1}\|_2$. In addition, when the equal sign in Theorem 1 holds, it can explain the assumption of regression metric loss (Chao *et al.*, 2022) that the distance between the features should be proportional to the distance between their corresponding labels. Specifically, Chao *et al.* (2022) uses a learnable parameter to restrain the proportion between feature distance and label distance. However, according to Theorem 1, this proportion should be related to the optimal weight $W_p^*$, and the equation may not hold when the labels are continuous. Moreover, representing the proportion with a constant ignores its fluctuations and variances among different samples. To alleviate this issue, we formulate this proportion as a mapping function and minimize its standard deviation to constrain the distance between the features to be uniform along the samples.

According to Theorem 1, the result of $d_r(\cdot, \cdot)$ should be a bounded proportion map and can be a constant function in some ideal situation. Hence, we minimize the standard deviation of $d_r(\cdot, \cdot)$ to acquire a flatter proportion map in a mini-batch. The loss function should be:

$$L_{std} = \sqrt{\frac{1}{N_b^2 - 1} \sum_i^{N_b} \sum_j^{N_b} (d_r(f_i, f_j) - \bar{d}_r)} \tag{2}$$

Here, $\bar{d}_r$ is a constant function equal to the mean of the relative distances in the batch and $N_b$ is the batch size. Clearly, $L_{std}$ constrains the predictor $p$ as a Lipschitz continuous function satisfying Remarks 1 and 2.

## 3.3 SPECTRAL ALIGNMENT OF DOMAINS

Existing works (Xu *et al.*, 2021; 2023) in domain generalization have demonstrated that the diversity and amount of training examples are positively correlated with the generalizability of a machine learning model. To expand the training set, we employ the data augmentation technique of c-mixup (Yao *et al.*, 2022) to generate additional samples from unseen distributions. However, without imposing a constraint of domain invariance, the learned feature space might include domain-specific information and thus become noisy (Liu *et al.*, 2023). This could hinder obtaining the optimal generalization power of the model.

To impose domain invariance constraint, the existing work of Chen *et al.* (2021) suggests not to minimize the difference between the Frobenius norm of feature representations of different domains, since the Frobenius norm may cause unstable performance. We assume that this instability can come from the fact that the Frobenius norm may encode the average of variances (i.e., singular values) along all orthogonal feature projections. We argue that the transferability of the feature representations mainly lies in aligning the highest variability directions corresponding to the largest

singular values Chen *et al.* (2019). Therefore, in our formulation, the Frobenius norm is substituted by the spectral norm, which only encodes the highest variability direction. We further show that the difference between spectral norms of features can be related to domain discrepancy.

**Notations** As Cortes & Mohri (2011), the expected loss in regression is $\mathcal{L}_D(h', h) = E_{x \sim D}[L(h(x), h'(x))]$ with $L$ being the MSE loss. We have the $\mathcal{L}_D(h, 0) = \frac{1}{N}\|\hat{Y}_D^h\|_F^2$, with $N$ being the number of samples, and $\hat{Y}_D^h$ being the output with hypothesis $h$ under distribution $D$. 0 represents the hypothesis mapping to zero element in $\mathcal{Y}$.

**Theorem 2.** *Given two distributions $P$ and $Q$, we have*

$$disc(P, Q) \leq \frac{1}{N} \max_{h \in H} |\|\hat{Y}_P^h\|_F^2 - \|\hat{Y}_Q^h\|_F^2|$$

*, where $disc$ represents the difference between distributions and $N$ denotes the number of the samples.*

**Proof 2.** *Generally speaking, we have*

$$\mathcal{L}(h', h) = \mathcal{L}(h - h', 0)$$

*Since $h, h'$ are in the subspace $H$ of Hilbert Space $\mathbb{H}$, we have $h'' = h - h' \in H$. Then, we have*

$$\forall h'' \in \mathbb{H}, disc(P, Q) \leq \max_{h'' \in H} |\mathcal{L}(h'', 0) - \mathcal{L}(h'', 0)|$$

*So, the proof is concluded.*

Theorem 2 shows the relation between the difference of feature representations and their distribution discrepancy. To determine the relation between the norm of the feature matrix and the output scale[1], we consider the spectral norm of the feature space, $\|F\|_2 = \sup_{w \neq 0} \frac{\|Fw\|_2}{\|w\|_2}$. If $W_i$ is a row vector of the weight $W$ in the fully connected layer, then $\|\hat{Y}_i^h\|_2 \leq \|\hat{Y}_i^h - b_i\|_2 + |b_i| \leq \|F\|_2\|W_i\|_2 + |b_i|$, $\hat{Y}_i^h$ is the $i$-th vector of the output matrix $\hat{Y}^h$ and $b_i$ is the $i$-th value of the bias vector $b$ in the fully connected layer. If we define $\lambda_i(F) = \|F\|_2\|W_i\|_2 + |b_i|$, we will have $\|\hat{Y}^h\|_F \leq \|\lambda(F)\|_2$.

From the discussion above, the spectral norm is related to the upper bound of the output scale. So aligning the spectral norms can prevent the output scales from differing greatly, which can also align two distributions as per Theorem 2. In this case, we propose a loss based on singular value decomposition (SVD) as follows:

$$L_{svd} = |max(s_{real}) - max(s_{syn})|, \tag{3}$$

where $s_{real}$ and $s_{syn}$ are the set of the singular values of the feature matrices from the real and synthesized distributions. The largest singular values of matrices are selected for calculating the loss. Note that $\|F\|_2 = max(s_F)$, where $s_F$ is the set of the singular values of matrix $F$.

### 3.4 OVERALL OBJECTIVE FUNCTION

We combine our objectives for relational contrastive learning and spectral alignment, and optimize them in an end-to-end training fashion. Formally, we have:

$$L = L_{mse} + \alpha L_{std} + \beta L_{svd}, \tag{4}$$

where $\alpha$ and $\beta$ represent hyper-parameters to balance the contribution of their corresponding loss functions. We further optimize the supervised loss of $L_{mse}$, formulated as:

$$L_{mse} = \frac{1}{N}(\sum_{i=1}^{N}(p(g(x_i^{real})) - y_i^{real})^2 + \sum_{i=1}^{N}(p(g(x_i^{syn})) - y_i^{syn})^2) \tag{5}$$

with $p(g(x_i^{real}))$ and $p(g(x_i^{syn}))$ being the prediction of input $x_i^{real}$ and the augmented sample $x_i^{syn}$, respectively. Here, $y_i^{real}$ and $y_i^{syn}$ denote the ground truth label corresponding to $x_i^{real}$ and $x_i^{syn}$ respectively .

---

[1]The Frobenius norm of the output $\|\hat{Y}_P^h\|_F$ represents the scale of the output in distribution $P$. Unlike classification, in regression, the target for each sample can be a vector. That means, if we have $N$ samples, each with $M$ dimensional target vectors, then $\hat{Y}_P^h$ is an $N \times M$ matrix.

## 4 EXPERIMENTAL RESULTS

### 4.1 IMPLEMENTATION DETAILS

Recent research (Kumar *et al.*, 2022; Kirichenko *et al.*, 2023) reveals a phenomenon that fine-tuning the whole network on a new task can improve the in-distribution (ID) performance of the new task, at the price of its out-of-distribution (OOD) accuracies. This is because fine-tuning the whole network changes the feature space spanned by the training data of a new task, which distorts the pretrained features. While linear probing can be an alternative solution to fine-tuning, due to its inability to adapt the features to the downstream task, it may degenerate the performance on in-distribution tasks. To mitigate this ID-OOD trade-off, motivated by the discussion in (Kumar *et al.*, 2022; Kirichenko *et al.*, 2023), we freeze the top of the C-mixup (Yao *et al.*, 2022) pretrained network (excluding the last block and the linear layers) during the training process. Specifically, we only fine-tune the bottom layer to preserve the low-level features from the pretrained model and unfreeze the last block to avoid degeneracy in the in-distribution tasks. In the following part, we use FT as an abbreviation for fine-tuning.

### 4.2 IN-DISTRIBUTION GENERALIZATION

**Datasets and experimental settings.** We evaluate the in-distribution (ID) generalization ability of our method on two tabular datasets (*i.e.*, Airfoil (Kooperberg, 1997), No2 (Kooperberg, 1997)), and one time-series dataset (*i.e.*, Exchange-Rate (Lai *et al.*, 2018a)). Airfoil contains 1503 data of aerodynamic and acoustic test results for different sizes of airfoil type NACA0012, while No2 is a collection of 500 data of air pollution related to traffic volume and meteorological variables. The Exchange-Rate dataset is a time-series dataset with a length of 7588, consisting of daily exchange rate data of eight countries from 1990 to 2016. Following Yao *et al.* (2022), we use a three-layer linear layer network for Airfoil and No2, and LST-Attn (Lai *et al.*, 2018b) for Exchange-rate. The preprocessing method on each dataset is the same as Yao *et al.* (2022). We also provide the result of RML (Chao *et al.*, 2022) combined with our fine-tuning method in the experiments of ID generalization. Additionally, we have conducted comparisons with the Feature Distribution Smoothing (FDS) method Yang *et al.* (2021), as well as with RankSim Gong *et al.* (2022). In it metric loss, RankSim considers the discrepancy between the order of feature distances and the order of label distances, rather than their proportion. Two evaluation metrics are considered for the performance on in-distribution tasks, namely Root Mean Square Error (RMSE) and Mean Averaged Percentage Error (MAPE). The results of our method and reproduced results are obtained by averaging three runs with different random seeds.

| | Airfoil | | No2 | | Exchange-Rate | |
|---|---|---|---|---|---|---|
| | RMSE↓ | MAPE (%)↓ | RMSE↓ | MAPE (%)↓ | RMSE↓ | MAPE (%)↓ |
| ERM† | 2.901 | 1.753 | 0.537 | 13.615 | 0.0236 | 2.423 |
| ERM* | 2.755 | 1.690 | 0.529 | 13.402 | 0.0257 | 2.613 |
| k-Mixup† (Greenewald *et al.*, 2021) | 2.938 | 1.769 | 0.519 | 13.173 | 0.0236 | 2.403 |
| Mixup† (Zhang *et al.*, 2018) | 3.730 | 2.327 | 0.528 | 13.534 | 0.0239 | 2.441 |
| Mani-Mixup† (Verma *et al.*, 2019) | 3.063 | 1.842 | 0.522 | 13.382 | 0.0242 | 2.475 |
| C-Mixup† (Yao *et al.*, 2022) | 2.717 | 1.610 | 0.509 | 12.998 | 0.0203 | 2.041 |
| C-Mxiup* | 2.736 | 1.639 | 0.516 | 13.069 | 0.0235 | 2.415 |
| FT | 2.541 | 1.474 | 0.519 | 13.201 | 0.0233 | 2.387 |
| FT+RML | 2.560 | 1.496 | 0.537 | 13.801 | 0.0179 | 1.838 |
| FT+RankSim | 2.635 | 1.537 | 0.520 | 13.188 | - | - |
| FT+FDS | 2.663 | 1.529 | 0.589 | 14.986 | 0.0235 | 2.397 |
| FT+$L_{std}$ | 2.586 | 1.501 | 0.510 | **12.879** | **0.0161** | **1.529** |
| FT+$L_{svd}$ | **2.489** | **1.443** | 0.517 | 13.161 | 0.0233 | 2.391 |
| FT+$L_{std}$+$L_{svd}$ | 2.516 | 1.460 | **0.506** | 12.896 | 0.0176 | 1.691 |

Table 1: Comparison on in-distribution datasets. The **bold** number is the best result and the underlined number is the second best result. The results of methods with † are reported by Yao *et al.* (2022) and the results of methods with * are reproduced based on the source code of Yao *et al.* (2022).

**Performance comparison.** We evaluate the ID generalization over three datasets and show the performance in Table 1. As we can see from the table, our method outperforms all the comparison methods in the ID generalization tasks. We find that the performance of $L_{std}$ outperforms RML in

most cases. As discussed above, RML only considers the scale of the proportion and ignores the variance which may not be able to have a better performance than $FT + L_{std}$. In addition, since the scales of the three datasets are not large enough, the pretrained model and our $FT + L_{svd}$ with synthesized distribution can also contribute to the improvement of in-distribution generalization on these datasets in some cases.

**t-SNE visualization** It is well known that traditional contrastive learning methods aim at learning compact feature clusters in the embedding space (Wen *et al.*, 2016; Schroff *et al.*, 2015). As aforementioned in the section on method, such clustering motivation may not be suitable for the regression tasks, but there are still connections between metric learning methods in regression and classification. According to our discussion, $L_{std}$ is trying to get a flatter $d_r$, which means the feature distribution should follow a discriminative pattern with less variance. To test the effect of contrastive learning in regression on embedding space, we visualize the feature distribution without metric loss, with RML, and with $L_{std}$ on Figure 4. This visualization can strongly support our assumption and discussion above. As Figure 4 shows, the feature distribution is more dispersed and the distribution pattern is clearer with $L_{std}$. In addition, as we discussed, RML focuses on learning a scale of the matrix feature and ignores the variance in the proportion. So, in some situations, the pattern will be blurred with RML, which is the same as the one shown in Figure 4. Note that $L_{std}$ maintains the property of being Lipschitz continuous for the predictor, which enhances the continuity of the feature distribution with less steep slopes. Figures 1c and 1d illustrate this difference: ulike $L_{std}$, RankSim Gong *et al.* (2022), which focuses solely on the distance between orders, does not preserve Lipschitz continuity. This characteristic might contribute to $L_{std}$'s superior performance over RankSim in most scenarios, as shown in Tables 1 and 2. It will also contribute to breakpoints in Figure 1c, which supports this hypothesis. Additional visualizations on $L_{svd}$ and $L_{std} + L_{svd}$ are provided in the Appendix.

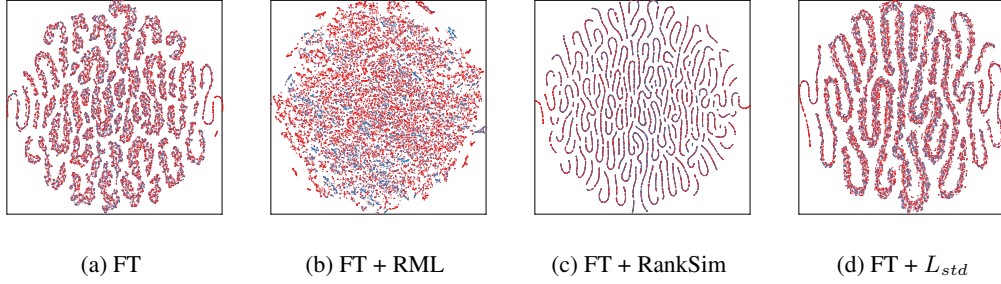

(a) FT      (b) FT + RML      (c) FT + RankSim      (d) FT + $L_{std}$

Figure 1: T-SNE visualization of the embedding space on DTI dataset. The visualizations from left to right are (a) The baseline model that is fine-tuned to minimize MSE loss, (b) The model that is fine-tuned to minimize both MSE and RML objectives, (c) the model that is finetuned to minimize both MSE and RankSim, and (d) The model that is fine-tuned to minimize both MSE and our developed $L_{std}$. The red points represent the features extracted from the train set and the blue points represent the features extracted from the test set. It is obvious that the pattern of the feature distribution is clearer with $L_{std}$.

### 4.3 OUT-OF-DISTRIBUTION GENERALIZATION

**Datasets** The out-of-distribution (OOD) generalization ability of models is evaluated over five datasets, including three real-world datasets (*i.e*, Communities&Crimes (Redmond, 2009), Skill-Craft (Mark Blair & Chen, 2013), Drug-targetInteractions (DTI) (Huang *et al.*, 2021)), one synthetic dataset (*i.e.*, RCF-MNIST (Yao *et al.*, 2022)), and one dataset contains both synthetic and real images (*i.e.,* MPI3D Gondal *et al.* (2019)). The Crimes and SkillCraft are two tabular datasets. The crimes dataset combines 1994 socio-economic data from three different sources and SkillCraft contains 3,395 video game telemetry data of real-time strategy (RTS) games from eight leagues. DTI is designed to predict the binding activity score between each small molecule and the corresponding target protein by collecting 232,458 data on the drug and target protein information. RCF-MNIST is a dataset with 60,000 images built on FashionMNIST (Xiao *et al.*, 2017) with spurious correlations between colours and rotation angles. MPI3D is a benchmark dataset of 1,036,800 images with three

distributions to predict intrinsic factors. In our experiments, we only consider the prediction of the rotation around a vertical and horizontal axis.

**Experimental settings** We evaluate our method on four datasets, namely RCF-MNIST, Crime, SkillCraft, and DTI. We leverage a three-layer linear layer network on Community&Crime and SkillCraft. Resnet18 (He *et al.*, 2016) is incorporated as the feature extractor for RCF-MNIST, and we employ DeepDTA (Öztürk *et al.*, 2018) on DTI.

Following the original paper of DTI (Huang *et al.*, 2021), we evaluate the methods on $R$ value. For the other three datasets, the evaluation metric is Root Mean Square Error (RMSE). When evaluating the out-of-distribution robustness, same as Yao *et al.* (2022), we report both average and worst-domain performance for the OOD experiments. Also, all the experiments are run over 3 seeds.

**Performance comparison.** The performance of OOD robustness on the four datasets is shown in Table 2. As the table shows, our method can achieve superior performance in most cases. For the datasets with small sizes, the pretained model plays an important role in improving generalization, since the scarcity of data is the key problem in these datasets. Also, the distribution alignment with $L_{svd}$ can enhance the OOD robustness as well. In addition, we find that $L_{std}$ also has the ability to generalize the spurious correlation as shown by the results of RCF-MNIST. We assume that the spurious correlation increases the variance in the proportion, which can be generalized by $L_{std}$.

| | RCF-MNIST | Crime | | SkillCraft | | DTI | |
|---|---|---|---|---|---|---|---|
| | RMSE↓ | RMSE↓ | | RMSE↓ | | $R$↑ | |
| | Avg. | Avg. | Worst | Avg. | Worst | Avg. | Worst |
| ERM† | 0.162 | 0.134 | 0.173 | 5.887 | 10.182 | 0.464 | 0.429 |
| ERM* | 0.160 | 0.135 | 0.172 | 6.151 | 7.916 | 0.475 | 0.438 |
| IRM† (Arjovsky *et al.*, 2019) † | 0.153 | **0.127** | 0.155 | 5.937 | 7.849 | 0.478 | 0.432 |
| IB-IRM† (Ahuja *et al.*, 2021) | 0.167 | **0.127** | **0.153** | 6.055 | 7.650 | 0.479 | 0.435 |
| CORAL† (Li *et al.*, 2018) | 0.163 | 0.133 | 0.166 | 6.353 | 8.272 | 0.483 | 0.432 |
| GroupDRO† (Sagawa *et al.*, 2019) | 0.232 | 0.138 | 0.168 | 6.155 | 8.131 | 0.442 | 0.407 |
| mixup† (Zhang *et al.*, 2018) | 0.176 | 0.128 | 0.154 | 5.764 | 9.206 | 0.465 | 0.437 |
| C-Mxiup* (Yao *et al.*, 2022) | 0.153 | 0.131 | 0.166 | 5.860 | 8.795 | 0.483 | 0.449 |
| FT | 0.146 | 0.129 | 0.156 | 5.592 | 8.358 | 0.479 | 0.458 |
| FT+RML | 0.167 | 0.129 | **0.153** | 5.496 | 8.249 | 0.480 | 0.446 |
| FT+RankSim | 0.239 | 0.135 | 0.164 | 5.324 | 7.577 | 0.479 | 0.464 |
| FT+FDS | 0.147 | 0.129 | 0.160 | **5.201** | **6.908** | 0.479 | 0.445 |
| FT+$L_{std}$ | **0.145** | 0.128 | 0.157 | 5.592 | 8.355 | **0.491** | **0.479** |
| FT+$L_{svd}$ | 0.147 | 0.129 | 0.159 | 5.591 | 8.358 | 0.479 | 0.444 |
| FT+$L_{std}$+$L_{svd}$ | 0.146 | **0.127** | 0.161 | 5.592 | 8.355 | 0.484 | 0.469 |

Table 2: Comparison on out-of-distribution datasets. The **bold** number is the best result and the underlined number is the second best. The results of methods with † are reported by Yao *et al.* (2022). The results of methods with * are reproduced based on the source code of Yao *et al.* (2022).

**Results on MPI3D dataset** We analyze our method under the setting of domain generalization on MPI3D dataset, which is a benchmark dataset for Domain Adaptation in Regression. We adapt a domain generalization settings (Gulrajani & Lopez-Paz, 2021) by evaluating our method over three generalization tasks on MPI3D dataset: **rl, rc → t**; **t, rc → rl**; **rl, t → rc**. We use the test sets of source distributions as the validation sets for the model selection. All the experiments are run over three random seeds, and we follow Cha *et al.* (2021) for random seed and hyper-parameter seed selection. The evaluation metrics on this task are Mean Square Error (MSE) and Mean Absolute Error (MAE). Since MPI3D is a large dataset containing 1,036,800 examples, we do not use our fine-tuning method on this dataset and there is no frozen parameter during the training process.

The MSE and MAE results are shown in Table 3 and 4 respectively. The comparison between $L_{std}$ and RML (Chao *et al.*, 2022) shows the advantage of regarding the proportion as a fluctuating map instead of a constant. In addition, the performance also shows that the alignment with $L_{svd}$ can significantly improve the generalization ability in some cases. We also provide the results of alignment with Nuclear-norm $\|\cdot\|_*$ and Frobenius norm $\|\cdot\|_F$. With norm equivalence (Cai *et al.*, 2016), $\|\cdot\|_2 \leq \|\cdot\|_F \leq \|\cdot\|_*$, the spectral norm can give a tighter upper bound. This can explain the reason that $L_{svd}$ can get the best performance among them.

| | MPI3D-MSE | | | |
|---|---|---|---|---|
| | rc | rl | t | Avg. |
| ERM | $0.08132 \pm 9.6e^{-6}$ | $0.09819 \pm 6.2e^{-5}$ | $0.007004 \pm 5.4e^{-9}$ | 0.06217 |
| C-Mixup | $0.09226 \pm 4.2e^{-5}$ | $0.10495 \pm 1.8e^{-4}$ | $0.014453 \pm 5.9e^{-8}$ | 0.07055 |
| RML | $0.08596 \pm 5.6e^{-5}$ | $0.09412 \pm 6.3e^{-6}$ | $0.020132 \pm 1.3e^{-6}$ | 0.06676 |
| Nuclear-norm | $0.09490 \pm 8.1e^{-5}$ | $0.09536 \pm 5.8e^{-4}$ | $0.011940 \pm 3.1e^{-6}$ | 0.06740 |
| F-norm | $0.09565 \pm 1.2e^{-5}$ | $0.10548 \pm 2.4e^{-2}$ | $0.008318 \pm 4.0e^{-6}$ | 0.06981 |
| $L_{std}$ | $0.07984 \pm 8.2e^{-5}$ | $0.09624 \pm 2.7e^{-5}$ | $0.006996 \pm 1.1e^{-8}$ | 0.06103 |
| $L_{svd}$ | $\mathbf{0.07942} \pm 4.9e^{-5}$ | $0.08355 \pm 1.1e^{-4}$ | $\mathbf{0.006016} \pm 1.3e^{-7}$ | 0.05633 |
| $L_{std} + L_{svd}$ | $0.07956 \pm 4.0e^{-5}$ | $\mathbf{0.07885} \pm 2.0e^{-5}$ | $0.006017 \pm 1.6e^{-7}$ | **0.05481** |

Table 3: Comparison on MPI3D dataset with the setting of domain generalization under the MSE index. The **bold** number is the best result. The unseen domains are labeled on the top.

| | MPI3D-MAE | | | |
|---|---|---|---|---|
| | rc | rl | t | Avg. |
| ERM | $0.3163 \pm 3.3e^{-5}$ | $0.3511 \pm 3.2e^{-4}$ | $0.0922 \pm 6.7e^{-7}$ | 0.2532 |
| C-Mixup | $0.3367 \pm 1.6e^{-4}$ | $0.3666 \pm 5.5e^{-4}$ | $0.1296 \pm 5.1e^{-6}$ | 0.2776 |
| RML | $0.3315 \pm 1.3e^{-4}$ | $0.3448 \pm 1.8e^{-5}$ | $0.1661 \pm 4.4e^{-5}$ | 0.2808 |
| Nuclear-norm | $0.3270 \pm 2.4e^{-4}$ | $0.3313 \pm 1.7e^{-3}$ | $0.1181 \pm 5.3e^{-5}$ | 0.2588 |
| F-norm | $0.3226 \pm 4.6e^{-5}$ | $0.3411 \pm 6.2e^{-3}$ | $0.0985 \pm 2.2e^{-5}$ | 0.2541 |
| $L_{std}$ | $0.3149 \pm 3.2e^{-4}$ | $0.3478 \pm 1.3e^{-4}$ | $0.0919 \pm 9.6e^{-7}$ | 0.2515 |
| $L_{svd}$ | $\mathbf{0.3016} \pm 9.8e^{-5}$ | $0.3225 \pm 5.0e^{-4}$ | $\mathbf{0.0856} \pm 1.1e^{-5}$ | 0.2366 |
| $L_{std} + L_{svd}$ | $0.3058 \pm 1.0e^{-4}$ | $\mathbf{0.3137} \pm 1.1e^{-4}$ | $0.0858 \pm 1.4e^{-4}$ | **0.2351** |

Table 4: Comparison on MPI3D dataset with the setting of domain generalization under MAE index. The **bold** number is the best result. The unseen domains are labeled on the top.

## 4.4 Hyper-parameter Sensitivity Analysis

We analyze the hyper-parameters on $\alpha$ and $\beta$ in Equation 4. Since the value of $L_{mse}$ is always much smaller than the value of $L_{std}$ and $L_{svd}$, we hope the two hyper-parameters can be smaller than 1. So, we analyze the trend of the performance of $L_{std}$ and $L_{svd}$ with $\alpha$ and $\beta$ in the range between $[1e^{-9}, 1e^4]$. Figure 2 shows the sensitivity of the hyper-parameters on in-distribution dataset No2 and out-of-distribution dataset DTI respectively. We find that the $L_{std}$ is much more sensitive since the value of $L_{std}$ is usually much larger than $L_{mse}$ and $L_{svd}$. More analysis of $\beta$ on MPI3D dataset is shown in the Appendix.

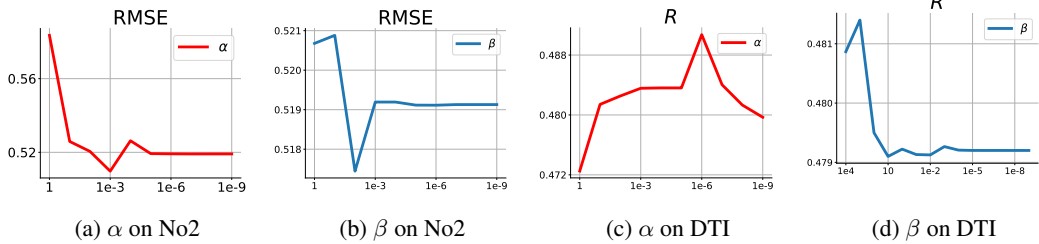

(a) $\alpha$ on No2          (b) $\beta$ on No2          (c) $\alpha$ on DTI          (d) $\beta$ on DTI

Figure 2: Hyper-parameter analysis on No2 and DTI datasets. For RMSE, the smaller value means the better result. For $R$, the larger value means the better result.

## 5 Conclusion

This paper discusses two main objectives that are required to improve generalization in regression. For In-Distribution generalization, we propose relational contrastive learning loss, based on the assumption that the distance between features and their corresponding labels should be correlated. We assume that the proportion between feature distance and label distance is a mapping function. Through this loss, we show that the variance in the embedding space is decreased, resulting in more discriminative patterns. To improve the transferability of the model on out-of-distribution data, we propose to augment the original data and then align the synthesized and real distributions through minimizing the difference between spectral norm of features.

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

APPENDIX

## A   T-SNE VISUALIZATION

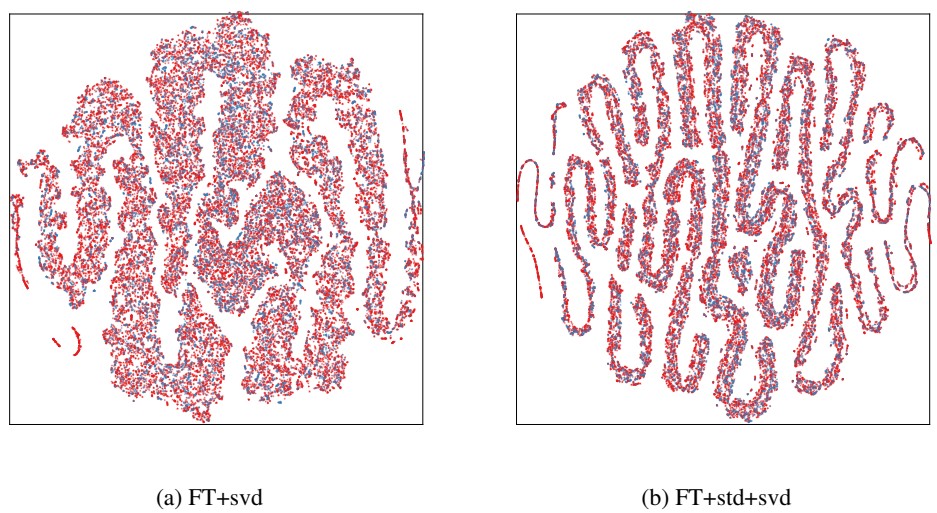

(a) FT+svd          (b) FT+std+svd

Figure 3: T-SNE visualization of the embedding space on DTI dataset with $L_{svd}$ and $L_{std}+L_{svd}$.

## B   HYPERPARAMETER ANALYSIS

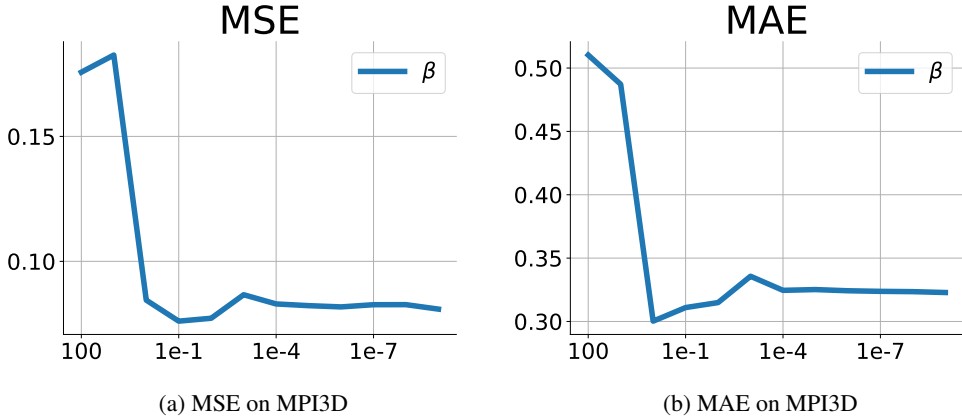

(a) MSE on MPI3D          (b) MAE on MPI3D

Figure 4: Analysis of $\beta$ on MPI3D when rl is the test domain

