# OpenReview forum: "Spectral Contrastive Regression"
_ICLR.cc/2024/Conference — Submitted to ICLR 2024_

### Official Review · Reviewer_xKM4 · 2023-10-27

**Soundness:** 3 good
**Presentation:** 2 fair
**Contribution:** 2 fair
**Rating:** 5
**Confidence:** 4

**Summary:**

This paper tackle the problem of generalization for regression tasks. It proposes a method based on metric learning assumption that the distance between features and labels should be proportional. defined as a mapping function. The proposed loss function aims at minimizing the error of the mapping function for the proportion and stabilizing its fluctuating behavior by smoothing out its variations. To enable out-of-distribution generalization, it also proposes to align the maximum singular value of the feature matrices across different domains. The paper conducts experiments on both in-distribution generalization and out-of-distribution robustness and shows that the proposed method can achieve superior performance in most cases.

**Strengths:**

The method is quite novel, and the empirical results are promising.

**Weaknesses:**

My main concern is that some related works / baselines are missing in this paper. It is not as the authors claimed that regression generalization remains relatively underexplored. Also, there are already many papers try to align the order of feature distances with the order of label distances, and they also evaluated OOD/zero-shot generalization, such as:

[1] Yang et al. Delving into Deep Imbalanced Regression. ICML 2021.

[2] Gong et al. RankSim: Ranking Similarity Regularization for Deep Imbalanced Regression. ICML 2022.

[3] Zha et al. Rank-N-Contrast: Learning Continuous Representations for Regression. NeurIPS 2023.

I think the authors should avoid claiming this paper introduces the contrastive interdependence between features and labels, and discuss about and compare with the above papers.

Minor: It would be better to give your method a name, instead of  FT+L_std+L_svd.

**Questions:**

1. Does FT refer to fine-tuning in the experiments? It's better to explain it in texts.

2. In the experiments, FT+L_std+L_svd seldom gets the best results over all FT methods. Is there an explanation?

---

> ### Author Response · Authors · 2023-11-15
>
> Thank you very much for your time and feedback. We incorporated your feedback in the updated paper by adding the results of RankSim and FDS to the tables. Specifically:
> 1. We report the results with the Feature Distribution Method (FDS) by conducting the experiments over 7 datasets. The results are shown in Tables 1 and 2 of the updated paper. As can be seen, our method outperforms FDS in most cases.
> 2. Unlike our approach, the paper "RankSim: Ranking Similarity Regularization for Deep Imbalanced Regression" only considers the relation between the order of feature distances and the order of label distances, and not their proportions. Therefore, it can not guarantee the Lipschitz continuous property of the predictor $p$. In other words, RankSim may lead to very steep inclines in the feature distribution, which may affect the continuity of the feature distribution. We add the results of RankSim in the tabels as well. Since the RankSim is only designed on the dataset with discrete target labels, we skip those datasets where RankSim can not be applied. The result is shown in Tables 1 and 2. Our method outperforms RankSim in most cases. Also, since RankSim has a similar motivation with $L_{std}$, we also visualize the resulting feature distribution on DTI dataset in Figure 1 of the updated paper, and compare it to our method. As can be seen, even though Ranksim has a more discriminative pattern, there are more breakpoints in the distribution because of lacking the Lipschitz continuous property. This may be the reason that our proposed $L_{std}$ can perform better than RankSim.
> 3. The paper "Delving into Deep Imbalanced Regression." proposed two strategies to smooth label distribution and feature distribution. Since the Label Distribution Smoothing (LDS) is applied on discrete target labels, it cannot be directly applied to our datasets where the labels have real/continues values. More details can be found at the github link https://github.com/YyzHarry/imbalanced-regression/blob/a6fdc45d45c04e6f5c40f43925bc66e580911084/agedb-dir/datasets.py#L60
>
>
>
> |                        | Airfoil        | Airfoil       | No2            | No2           | ExchangeRate    | ExchangeRate   |
> |:----------------------:|:--------------:|:-------------:|:--------------:|:-------------:|:---------------:|:--------------:|
> |                        | RMSE  $\downarrow$          | MAPE(%)  $\downarrow$     | RMSE $\downarrow$           | MAPE(%)$\downarrow$       | RMSE$\downarrow$             | MAPE(%) $\downarrow$       |
> | FT+RankSim             | 2.635          | 1.537         | 0.520          | 13.188        | -               | -              |
> | FT+FDS                 | 2.663          | 1.529         | 0.589          | 14.986        | 0.0235          | 2.397          |
> | FT                     | 2.541          | 1.474         | 0.519          | 13.201        | 0.0233          | 2.387          |
> | FT+$L_{std}$           | 2.586        | 1.501         | 0.510          | 12.879 | 0.0161 | 1.529 |
> | FT+$L_{svd}$           | 2.489 | 1.443 | 0.517          | 13.161        | 0.0233          | 2.391          |
> | FT+$L_{std}$+$L_{svd}$ | 2.516          | 1.460         | 0.506 | 12.896        | 0.0176          | 1.691          |
>
>
> |                        | RCF-MNIST             | Crime                 | Crime                  | SkillCraft            | SkillCraft             | DTI         | DTI               |
> |:----------------------:|:---------------------:|:---------------------:|:----------------------:|:---------------------:|:----------------------:|:-----------:|:-----------------:|
> |                        | avg. RMSE$\downarrow$ | avg. RMSE$\downarrow$ | worst RMSE$\downarrow$ | avg. RMSE$\downarrow$ | worst RMSE$\downarrow$ | R$\uparrow$ | worst R$\uparrow$ |
> | FT+RankSim             | 0.239                 | 0.135                 | 0.164                  | 5.324                 | 7.577                  | 0.479       | 0.464             |
> | FT+FDS                 | 0.147                 | 0.129                 | 0.160                  | 5.201                 | 6.908                  | 0.479       | 0.445             |
> | FT                     | 0.146                 | 0.129                 | 0.156                  | 5.592                 | 8.358                  | 0.479       | 0.458             |
> | FT+$L_{std}$           | 0.145                 | 0.128                 | 0.157                  | 5.592                 | 8.355                  | 0.491       | 0.479             |
> | FT+$L_{svd}$           | 0.147                 | 0.129                 | 0.159                  | 5.591                 | 8.358                  | 0.479       | 0.444             |
> | FT+$L_{std}$+$L_{svd}$ | 0.146                 | 0.127                 | 0.161                  | 5.592                 | 8.355                  | 0.484       | 0.469             |

---

> ### Author Response · Authors · 2023-11-15
>
> 4. We note that the paper 'Rank-N-Contrast: Learning Continuous Representations for Regression' was published on Arxiv after the submission of our paper and remains unpublished. Given its post-submission release and unpublished status, it was not valid for comparison in our study.
>
> **Questions**
>
> Q1: FT stands for Fine-tuning; we included this explanation in Section 4.1 of the revised paper.
>
> Q2: $L_{std}$ demonstrates better performance over the baselines in numerous instances. Please note that the mixup technique synthesizes the data with a new label distribution. When the new label distribution differs significantly from the original, the $L_{svd}$ loss might mislead the model into aligning the label distributions by constraining the scale of the output. We hypothesize that high label and sample density can prevent this issue, a condition that holds true in the presence of large datasets. Therefore, given the large size of MPI3D, $L_{svd}$ is more suitable for this dataset compared to the other seven, as detailed in Table 3 of our updated paper.

---

### Official Review · Reviewer_xHdj · 2023-10-29

**Soundness:** 3 good
**Presentation:** 2 fair
**Contribution:** 3 good
**Rating:** 5
**Confidence:** 3

**Summary:**

This paper presents an innovative approach for generalizing regression tasks by leveraging the metric learning assumption that emphasizes the proportional relationship between features and labels. The method incorporates a std. loss and spectral loss to address two key aspects: ensuring the distance proportionality between features and labels and enabling OOD generalization. The effectiveness of the proposed method is demonstrated through experiments conducted on multiple datasets.

**Strengths:**

1. This work addresses an emerging issue in regression tasks, namely the challenge of handling OOD data. As the author notes, while OOD generalization has been studied in classification tasks, it has not been explored in depth for regression tasks.

2. The proposed method involves measuring the feature-label distance proportion using a mapping function and aligning real and synthesized distributions by minimizing the difference between the spectral norms of their feature representations.

**Weaknesses:**

1. The organization and statements in this paper can be unclear at times, as the authors attempt to cover a lot of ground on the topic. For example, the abstract section contains too many details that may not be necessary. In essence, the paper proposes an OOD generalization method for regression tasks, which involves two penalties to address feature-label distance proportion and distribution gap issues. However, some of the irrelevant expressions can make it difficult to grasp the main topic at first.

2. The title of the paper is also unclear and does not directly convey the main theme, similar to the abstract. It lacks a clear focus and fails to capture the essence of the research.

**Questions:**

1. I am confused about why the title is "Spectral Contrastive Regression." On one hand, the title does not explicitly mention OOD generalization, which is the main focus of the paper. On the other hand, the term "spectral" does not seem to directly relate to the contrastive loss used in the paper. While the paper introduces concepts of spectral and contrastive learning in the context of a regression task, the title may give the impression of avoiding the core content and introducing the concept of contrastive learning.

2. Throughout the entire paper, the concept of contrastive learning is not emphasized enough. The term "contrastive" appears only five times in the main text and is not even mentioned in the abstract. While this expression may not be crucial for the technical contributions of the paper, the overall writing style feels somewhat disjointed. Unlike traditional contrastive learning, the concept is not reinforced, and even after reading about the std. loss, it is surprising to see the section titled "Relational Contrastive Learning." The paper gives the impression of being written in a fragmented manner. This is just my personal perception and may not necessarily be correct.

3. Building upon the previous point, I understand that the authors utilize the relationship between feature and label distances, adopting a contrasting perspective to examine this proportion and control its fluctuation by proposing a loss based on standard deviation. However, I still question why this loss and the keyword "contrastive" are not aligned, and instead, the paper introduces the concepts of standard deviation and the corresponding expressions in the abstract. Overall, it might be my personal bias, but I feel that the writing in this paper lacks cohesion.

4. Regarding the issue with the loss function, in Equation 4, both the first and third terms measure the difference between two distributions. The former considers the MSE between the individual in-distribution of the two distributions, while the latter measures the difference between the two distributions themselves. However, it is unclear which distribution the third term specifically refers to. Does it pertain only to the real distribution or both distributions? Equation 2 appears to be a general constraint without specifying the source of i and j from each distribution.

5. The related work section appears to be somewhat perfunctory, as there is not much informative content provided in the three paragraphs.

---

> ### Author Response · Authors · 2023-11-22
>
> - The traditional contrastive learning in classification aims to generate a more discriminative feature distribution using contrastive pairs, namely positive pairs and negative pairs. Many contrastive losses, such as triplet loss, belong to the area of metric learning and aim at minimizing the distance between positive pairs and maximizing the distance between negative pairs. In the context of regression tasks, the $L_{std}$ loss can be considered a form of metric loss, as it introduces a new metric distance $d_r$. This adaptation of the contrastive concept to regression is due to the continuous nature of the labels.
>
>
> - Generating a discriminative distribution with continuous labels is a challenge in regression tasks. As discussed in the paper, the ideal situation is to have $d_r$ constant at $d_r^\ast = |{W^\ast}^{-1}_p|$. Under this framework, we can define two contrastive pair sets:
>
> $
>     S_l = \left[(f_i, f_j)|d_r(f_i,f_j) < d_r^\ast\right];
>     S_g = \left[{(f_i, f_j)|d_r(f_i,f_j) > d_r^\ast\}\right]
> $
>
>
> In this case, contrastive metric learning in regression aims to minimize the variance, ensuring that the feature distance
>  aligns more closely with the optimal proportion. This is achieved by minimizing the distance within $S_l$ (where feature pairs are closer than the ideal distance) and maximizing the distance within $S_g$ (where feature pairs are farther than the ideal distance). The $L_{std}$ loss helps to decrease unexpected variance in the feature space, thus maintaining the proportionality essential in regression tasks.
>
> - We thank the reviewer for the suggestion and acknowledge the need for a new title to better reflect the paper's content and to distinguish it from traditional concepts of contrastive learning. We propose an updated title of "Unifying Scale and Proportion for Feature-Level Generalization in Regression", and will replace the subtitle of Section 3.2 with: "Proportional distance for Continuous Label Spaces". These changes aim to more accurately convey the focus of the paper on optimizing feature distances in regression tasks and its contribution to metric learning with continuous labels.
>
> - The second term $L_{std}$ is calculated for the two distributions separately, represented as $L_{std}^{real}$ and $L_{std}^{syn}$, respectively. The final $L_{std}$ loss is the sum of $L_{std}^{real}$ and $L_{std}^{syn}$. The third term $L_{svd}$ is only applicable when there is more than one distribution. According to Equation 3, $L_{svd} = |max(s_{real}) - max(s_{syn})|$, where each singular value is calculated from one distribution. Therefore, there should be at least two distributions for the $L_{svd}$ term to be applicable.

---

### Official Review · Reviewer_Rgn5 · 2023-10-30

**Soundness:** 3 good
**Presentation:** 3 good
**Contribution:** 2 fair
**Rating:** 5
**Confidence:** 4

**Summary:**

To improve the generalization of deep regression problems, the authors present a new objective composed of several ideas including relational contrastive learning, spectral alignment, and augmented sample pairs. The experiments are extensively conducted on multiple benchmarks and show improvement over baselines.

**Strengths:**

- The manuscript is clear and easy to follow.
- The experiments show good results and are conducted on multiple different datasets.
- The idea is technically sound and the authors present a neat combination of several different ideas to improve the generalization of deep regression problems.

**Weaknesses:**

I'm concerned about the technical novelty. Though the experiments show good improvement over baselines, in the current state of the manuscript, the proposed objective is the combination of several different terms that are similar to some existing work. Please see point 1 and point 2 in the next section of Questions. The authors may consider including more ablation studies to further solidify the technical contribution.

**Questions:**

- More ablation studies on Eq.4. The authors have conducted ablation studies on $\alpha$ and $\beta$.

  In Eq.4, does $\mathcal{L}_{std}$ include augmented samples?

   Since $\mathcal{L}_{mse}$ includes the augmented samples, I suggest the authors also conduct an ablation study on how much improvement is introduced by using augmented samples in mse loss term.

- Missing related work. One of the core ideas in the proposed objective $\mathcal{L}_{std}$ is that the distance between features and labels should be proportional. A similar idea can be found in deep regression problems [1], which showed similar patterns in the feature space with t-SNE visualization. The authors should properly discuss and compare the differences and similarities.

- In Fig.2, $\beta$ introduces little to no effect on the metrics for varying its values in the entire range. Do the authors have any speculation or analysis on this pattern? Because from Tables 1 and 2, single svd loss term can provide significant improvement and sometimes has the best performance. But when combined with std, it does not show a significant effect.

- It can provide a full picture of how the proposed objective works if the authors can have t-SNE visualization for only svd loss term, and the sum of svd + std loss terms.




- Minor points: it might be better for the audience if the abbreviated 'FT' can be explained as 'fine-tuning' before using it.



[1] Gong et al., RankSim: Ranking Similarity Regularization for Deep Imbalanced Regression. ICML 2022

---

> ### Author Response · Authors · 2023-11-22
>
> Thank you very much for your time. We incorporated your feedback into the updated paper by adding more experiments. Additionally, we address your questions below.
>
> Q1. $L_{std}$ is calculated for both the original distribution and the augmented distribution. As requested by the reviewer, we have conducted extra experiments to study the effect of augmentation and the proposed $L_{std}$. The following table shows the results of an ablation study on $L_{std}$, with and without augmentation.
> |                           | No2              | No2                   | ExchangeRate     | ExchangeRate          | RCF-MNIST             | DTI                | DTI                 |           |
> |---------------------------|------------------|-----------------------|------------------|-----------------------|-----------------------|--------------------|---------------------|-----------|
> |                           | RMSE$\downarrow$ | MAPE (\%)$\downarrow$ | RMSE$\downarrow$ | MAPE (\%)$\downarrow$ | avg. RMSE$\downarrow$ | avg. $R$$\uparrow$ | worst $R$$\uparrow$ | avg. rate |
> | FT(w/o augment)           | 0.521            | 13.256                | 0.0236           | 2.405                 | 0.147                 | 0.479              | 0.444               | -         |
> | FT(w augment)             | 0.519(0.38\%)    | 13.201(0.45\%)        | 0.0233(1.27\%)   | 2.387(0.75\%)         | 0.146(0.68\%)         | 0.479(0\%)         | 0.458(3.15\%)       | 0.95\%    |
> | FT+$L_{std}$(w/o augment) | 0.517            | 13.120                | 0.0165           | 1.563                 | 0.152                 | 0.483              | 0.449               | -         |
> | FT+$L_{std}$(w augment)   | 0.510(1.35\%)    | 12.879(1.84\%)        | 0.0161(2.42\%)   | 1.529(2.18\%)         | 0.145(4.61\%)         | 0.491(1.66\%)      | 0.479(6.68\%)       | 2.96\%    |
>
> As can be seen from the table, FT+$L_{std}$(w augment) always outperforms FT(w/o augment). Compared with FT(w augment), FT+$L_{std}$(w augment) brings extra 1\%-3\% improvement on average.

---

> ### Author Response · Authors · 2023-11-22
>
> Q2. There are significant differences between Ranksim and our method. RankSim considers the relation between the order of feature distance and the order of label distance instead of the proportion. So, it is hard for RankSim to keep the predictor Lipschitz continuous. That can lead to steep slope in the feature distribution, which is shown by the discontinuity in the visualization of Figure 1 of the updated paper. The following tables show the results of RankSim compared with our method. We skip the datasets RankSim can not be applied. Our method outperforms RankSim in most cases. Also, since RankSim has a similar motivation with $L_{std}$, we also visualize the resulting feature distribution on DTI dataset in Figure 1 of the updated paper, and compare it to our method. As can be seen, even though Ranksim has a more discriminative pattern, there are more breakpoints in the distribution because of lacking the Lipschitz continuous property. This may be the reason that our proposed $L_{std}$ can perform better than RankSim.
>
> |                        | Airfoil        | Airfoil       | No2            | No2           | ExchangeRate    | ExchangeRate   |
> |:----------------------:|:--------------:|:-------------:|:--------------:|:-------------:|:---------------:|:--------------:|
> |                        | RMSE  $\downarrow$          | MAPE(%)  $\downarrow$     | RMSE $\downarrow$           | MAPE(%)$\downarrow$       | RMSE$\downarrow$             | MAPE(%) $\downarrow$       |
> | FT+RankSim             | 2.635          | 1.537         | 0.520          | 13.188        | -               | -              |
> | FT                     | 2.541          | 1.474         | 0.519          | 13.201        | 0.0233          | 2.387          |
> | FT+$L_{std}$           | 2.586        | 1.501         | 0.510          | 12.879 | 0.0161 | 1.529 |
> | FT+$L_{svd}$           | 2.489 | 1.443 | 0.517          | 13.161        | 0.0233          | 2.391          |
> | FT+$L_{std}$+$L_{svd}$ | 2.516          | 1.460         | 0.506 | 12.896        | 0.0176          | 1.691          |
>
>
> |                        | RCF-MNIST             | Crime                 | Crime                  | SkillCraft            | SkillCraft             | DTI         | DTI               |
> |:----------------------:|:---------------------:|:---------------------:|:----------------------:|:---------------------:|:----------------------:|:-----------:|:-----------------:|
> |                        | avg. RMSE$\downarrow$ | avg. RMSE$\downarrow$ | worst RMSE$\downarrow$ | avg. RMSE$\downarrow$ | worst RMSE$\downarrow$ | R$\uparrow$ | worst R$\uparrow$ |
> | FT+RankSim             | 0.239                 | 0.135                 | 0.164                  | 5.324                 | 7.577                  | 0.479       | 0.464             |
> | FT                     | 0.146                 | 0.129                 | 0.156                  | 5.592                 | 8.358                  | 0.479       | 0.458             |
> | FT+$L_{std}$           | 0.145                 | 0.128                 | 0.157                  | 5.592                 | 8.355                  | 0.491       | 0.479             |
> | FT+$L_{svd}$           | 0.147                 | 0.129                 | 0.159                  | 5.591                 | 8.358                  | 0.479       | 0.444             |
> | FT+$L_{std}$+$L_{svd}$ | 0.146                 | 0.127                 | 0.161                  | 5.592                 | 8.355                  | 0.484       | 0.469             |

---

> ### Author Response · Authors · 2023-11-22
>
> Q3.1 **``Because from Tables 1 and 2, single svd loss term can provide significant improvement and sometimes has the best performance. But when combined with std, it does not show a significant effect"**
>
> We are uncertain about your comment in Q3.1, as $L_{std}$ demonstrates better performance over the baselines in numerous instances. Please note that the mixup technique synthesizes the data with a new label distribution. When the new label distribution differs significantly from the original, the $L_{svd}$ loss might mislead the model into aligning the label distributions by constraining the scale of the output. We hypothesize that high label and sample density can prevent this issue, a condition that holds true in the presence of large datasets. Therefore, given the large size of MPI3D, $L_{svd}$ is more suitable for this dataset compared to the other seven, as detailed in Table 3 of our paper.
>
> Q3.2  **``In Fig.2, $\beta$ introduces little to no effect on the metrics for varying its values in the entire range. Do the authors have any speculation or analysis on this pattern? "**
>
> We agree that the impact on $L_{svd}$ appears insignificant in Fig. 2, likely due to the large scale of $L_{std}$. This might give the impression that $L_{svd}$ is unaffected by $\beta$. To clarify this, we have presented the hyperparameter analyses of $\alpha$ and $\beta$ separately in the revised version of our paper. For the analysis of $\beta$, we find that setting the hyperparameter to 1000 on the DTI dataset results in better performance for $L_{svd}$. As we discussed earlier, the MPI3D dataset might be more suited for $L_{svd}$, so we have also included a sensitivity analysis of $\beta$ for the MPI3D dataset in the Appendix of the updated paper. The impact of $L_{svd}$ is significant in some cases: the model achieves the best results when $\beta$ is set around 1. However, if $\beta$ is too large, causing $L_{svd}$ to dominate the total loss, the performance worsens
>
> Q4. We added the t-SNE visualization in the Appendix of the updated paper.
>
> Q5. FT stands for Fine-tuning; we included this explanation in Section 4.1 of the revised paper.

---

### Meta-Review · Area_Chair_bZ2e · 2023-12-07

**Metareview:**

Thanks for your submission to ICLR.

This paper considers an approach to constrastive / metric learning applied to regression.

This was a borderline paper and could have gone either way.  On the positive side, the reviewers noted that the paper appears to be technically sound, and some reviewers felt there was sufficient novelty.  On the negative side, other reviewers felt the novelty was not sufficient, that the organization of the paper could be improved, and that there were some missing baselines.  The authors provided a rebuttal, but this did not change the opinions of the reviewers.

I do think the rebuttal helped with some of the concerns of the paper, but at the end of the day, none of the reviewers were sufficiently positive about the paper to recommend acceptance.  It seems that at the least another version of the paper should be prepared that incorporates some of the suggestions made by the reviewers.

**Justification For Why Not Higher Score:**

Ultimately, none of the reviewers advocated for accepting the paper.

**Justification For Why Not Lower Score:**

N/A

---

### Decision · Program_Chairs · 2024-01-16

Reject